# Quantum Chaos, Randomness and Universal Scaling of Entanglement in Various Krylov Spaces

Hai-Long Shi[1⋆], Augusto Smerzi[1†] and Luca Pezzè[1‡]

**1** QSTAR, INO-CNR and LENS, Largo Enrico Fermi 2, 50125 Firenze, Italy

⋆ hailong.shi@ino.cnr.it
† augusto.smerzi@ino.cnr.it , ‡ luca.pezze@ino.cnr.it

## Abstract

Multipartite entanglement is a crucial resource for advancing quantum technologies, with considerable research efforts directed toward achieving its rapid and scalable generation. In this work, we derive an analytical expression for the time-averaged quantum Fisher information (QFI), enabling the detection of scalable multipartite entanglement dynamically generated by all quantum chaotic systems governed by Dyson's ensembles. Our approach integrates concepts of randomness and quantum chaos, demonstrating that the QFI is universally determined by the structure and dimension of the Krylov space that confines the chaotic dynamics. In particular, the QFI ranges from $N^2/3$ for $N$ qubits in the permutation-symmetric subspace (e.g. for chaotic kicked top models with long-range interactions), to $N$ when the dynamics extend over the full Hilbert space with or without bit reversal symmetry or parity symmetry (e.g. in chaotic models with short-range Ising-like interactions). In the former case, the QFI reveals multipartite entanglement among $N/3$ qubits and highlights the power of chaotic collective spin systems in generating scalable multipartite entanglement. Interestingly this result can be related to isotropic substructures in the Wigner distribution of chaotic states and demonstrates the efficacy of quantum chaos for Heisenberg-scaling quantum metrology. Finally, our general expression for the QFI agrees with that obtained for random states and, differently from out-of-time-order-correlators, it can also distinguish chaotic from integrable unstable spin dynamics.

## 1   Introduction

Engineering large multipartite entanglement in ensembles of many qubits is central to quantum simulations [1], information theory [2], and metrology [3]. These states are also pivotal for enhancing our understanding of condensed matter [4, 5] and high-energy physics [6, 7]. One of the primary objectives is the rapid generation of multipartite entanglement from readily available non-entangled states, such as (Gaussian) coherent spin states [8], using many-body dynamics [9–14].

The quantum Fisher information (QFI) not only quantifies the ultimate quantum sensing ability but also serves as a powerful detector of multipartite entanglement [15–17] in non-Gaussian states [18, 19]. The established connection [20–22] between out-of-time-order correlators (OTOCs)—a measure of information scrambling—and the QFI expands the scope of chaotic quantum systems [23] expected to exhibit exponentially fast generation of multipartite entanglement, owing to their ability to scramble information at an exponential rate. However, due to another constraint imposed by the Lieb-Robinson bounds on operator dynamics [24, 25], this does not generally imply that scrambling dynamics leads to scalable multipartite entanglement [26–28] in the long run. It thus remains unclear which types of chaotic systems can generate scalable multipartite entanglement, i.e., QFI scaling as $N^2$ known as the Heisenberg scaling.

In this Letter, we investigate multipartite entanglement generated by chaotic quantum dynamics for times longer than the scrambling (or Ehrenfest) time $t^*$. In particular, we study the time-averaged QFI,

$$\bar{F}_{\text{chaos}}[\hat{O}] \equiv \lim_{T \to \infty} \frac{1}{T - t^*} \int_{t^*}^{T} dt\, F_Q[|\psi_{\text{chaos}}(t)\rangle, \hat{O}], \tag{1}$$

where $F_Q[|\psi_{\text{chaos}}(t)\rangle, \hat{O}]$ is the QFI of the quantum state $|\psi_{\text{chaos}}(t)\rangle$ evolved under chaotic dynamics and $\hat{O}$ is a generic Hermitian operator. We compute analytically Eq. (1) based on standard random matrix theory (RMT) for all chaotic models described by Dyson's three circular ensembles [29–31] and using the ergodicity hypothesis [32, 33]. We demonstrate the universal behavior of QFI in chaotic systems

$$\bar{F}_{\text{chaos}}[\hat{O}] = \frac{4\text{Tr}_{\mathcal{K}}[\hat{O}^2]}{K} - \frac{4\text{Tr}_{\mathcal{K}}[\hat{O}]^2}{K^2} + \mathcal{O}\left(\frac{\text{Tr}_{\mathcal{K}}[\hat{O}^2]}{K^2}\right), \tag{2}$$

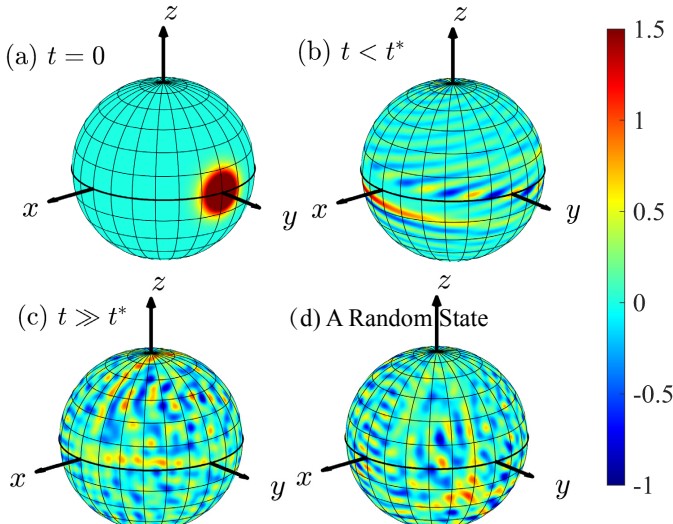

Figure 1: Panels (a)-(c) show the Wigner distributions (colormap) of the evolved state $|\psi_{\text{chaos}}(t)\rangle$ for the COE chaotic model Eq. (B.1): (a) $t = 0$, corresponding to the initial coherent spin state aligned along the $y$ axis; (b) $t < t^*$; and (c) $t \gg t^*$. Panel (d) shows the Wigner distribution of a typical example of random state with QFI close to $\bar{F}_{\text{rand}}$, Eq. (3).

which is universally governed by the structure and dimension $K \equiv \dim\mathcal{K}$ of the Krylov space $\mathcal{K}$. The Krylov space is the minimal subspace to confine the chaotic dynamics [34]. Although Eq. (2) holds for arbitrary $\hat{O}$, we mainly consider the special case $\hat{O} = J_\alpha$, where $J_\alpha = \sum_{j=1}^{N} \sigma_\alpha^{(j)}/2$ ($\alpha = x, y, z$) are collective spin operators of $N$ qubits, with $\sigma_\alpha^{(j)}$ being Pauli matrices. It predicts $\bar{F}_{\text{chaos}} \simeq N$ when the chaotic dynamics extends over the full $N$-qubit space $\mathcal{K} = \otimes^N \mathbb{C}^2$ ($K = 2^N$) or the subspace subjected to bit reversal symmetry or parity symmetry, which explains why only the linear scaling of QFI can be obtained in a chaotic Ising model. Instead, we find $\bar{F}_{\text{chaos}} \simeq N^2/3$ for the permutation-symmetric subspace $\mathcal{K} = \text{Sym}^N(\mathbb{C}^2)$ ($K = N+1$), as numerically confirmed by various kicked-top models. In this case, our results indicate that chaotic collective spin systems [35] can generate chaotic permutation-symmetric $N$-qubit states which exhibit at least $\lfloor N/3 \rfloor$-particle scalable entanglement. Furthermore, for permutation-symmetric states, we clarify that the universal scaling of QFI in Eq. (2) and its independence from spin direction $\alpha$ can distinguish chaotic from integrable collective spin systems with positive Lyapunov exponents (LEs) [21, 22, 36], in contrast to OTOCs.

Equation (2) has several remarkable consequences [37]. First, it provides a unifying view of randomness and quantum chaos, as expressed by the equality

$$\bar{F}_{\text{chaos}}[\hat{O}] \simeq \bar{F}_{\text{rand}}[\hat{O}] \equiv \int d\mu(U) F_Q[U|\phi_0\rangle, \hat{O}], \tag{3}$$

up to the leading terms in $K$, where the right-hand side quantity is the QFI averaged over random states in the space $\mathcal{K}$. Reference [38] proved that $\bar{F}_{\text{rand}}[\hat{O}] = 4\text{Tr}_{\mathcal{K}}[\hat{O}^2]/(K + 1) - 4(\text{Tr}_{\mathcal{K}}[\hat{O}])^2/K(K + 1)$, which agrees with Eq. (2) for $K \gg 1$ [39]. Notice that the unitary operators generating chaotic states are not Haar random [40]. Actually, the implementation of random unitary operators in a quantum system is exponentially hard in general [41–43]. Our findings demonstrate that randomness displayed by the QFI in Eq. (3) can be achieved in chaotic $N$-qubit systems involving only two-body interactions and initially polarized states, realizable in current atomic quantum simulators [44–53].

Furthermore, interesting insights can be gained by using the fundamental relation between

69 the QFI and the fidelity OTOC, $\mathcal{F}_{\hat{O}}(\theta,t) \equiv \langle\psi_0|W_{\hat{O}}^{\dagger}(\theta,t)\rho^{\dagger}(0)W_{\hat{O}}(\theta,t)\rho(0)|\psi_0\rangle$ [20–22,54]:

$$-2\frac{\partial^2 \mathcal{F}_{\hat{O}}(\theta,t)}{\partial\theta^2} = F_Q[|\psi_{\text{chaos}}(t)\rangle,\hat{O}], \tag{4}$$

70 where $\rho(0)=|\psi_0\rangle\langle\psi_0|$ is the initial pure state and $W_{\hat{O}}(\theta,t)=U_{\text{chaos}}^{\dagger}(t)\exp(i\theta\hat{O})U_{\text{chaos}}(t)$ [55].
71 Let us consider chaotic states in the permutation-symmetric subspace and use $\hat{O}=J_{\alpha}$. In this
72 case, Eqs. (2) and (4) predict that typically (namely, by replacing $F_Q[|\psi_{\text{chaos}}(t)\rangle,J_{\alpha}]$ for $t\gg t^*$
73 by its time average $\bar{F}_{\text{chaos}}[J_{\alpha}]$) the fidelity decreases sharply after a collective spin rotation
74 $\exp(i\delta\theta\hat{J}_{\alpha})$ by an angle $\delta\theta\sim\sqrt{3}/N$ around any axis $\alpha$: an evident non-Gaussian feature.
75 Consequently, we anticipate that chaotic states exhibit uniform angular substructures in the
76 generalized Bloch sphere, with a characteristic angular size $\delta\theta$. This is confirmed by plot-
77 ting the Wigner distribution of chaotic states obtained from a kicked top model (as discussed
78 below) and illustrated in Fig. 1(a)-(c). Starting from a coherent spin state [panel (a)], the
79 Wigner distribution of the chaotic state initially develops elongated structures for $t<t^*$ [panel
80 (b)]. Subsequently, for $t\gg t^*$, we observe characteristic isotropic substructures [panel (c)],
81 as expected, which indicates a high metrological sensitivity [56,57]. For comparison, panel
82 (d) displays the Wigner distribution of a random state with QFI close to the average given
83 by Eq. (3). The similarity between the Wigner distribution of $|\psi_{\text{chaos}}(t\gg t^*)\rangle$ with that of a
84 random state visually supports the results discussed above.
85   In the following, we detail the derivation and consequences of Eq. (2), also see Appendix,
86 and present the results of numerical simulations.

## 87 2  QFI for quantum chaotic dynamics

88 First, let us briefly recall the fundamental relationship between QFI and entanglement [3,58].
89 For a generic pure $N$-qubit state $|\psi\rangle\langle\psi|$, the QFI [59] is given by

$$F_Q[|\psi\rangle,\hat{O}] = 4(\langle\psi|\hat{O}^2|\psi\rangle - \langle\psi|\hat{O}|\psi\rangle^2). \tag{5}$$

90 Violation of the inequality $F_Q[|\psi\rangle,J_{\alpha}]\leq sk^2+r^2$, for $\hat{O}=J_{\alpha}$, indicates that $|\psi\rangle$ contains at least
91 $(k+1)$-particle entanglement among the $N$ qubits [15–17], where $s=\lfloor N/k\rfloor$ represents the
92 largest integer less than or equal to $N/k$, and $r=N-sk$. QFI can also be used to detect insepa-
93 rability and entanglement depth [60,61]. Next, we will derive the QFI for chaotic systems by
94 computing Eq. (5) using RMT and the ergodicity hypothesis.
95   The chaotic systems under consideration exhibit energy spectrum structures that, although
96 potentially belonging to distinct symmetry classes, are effectively modeled by Dyson's three
97 circular ensembles of unitary random matrices [29–31,62,63], which validates the reliability
98 of employing RMT. These are the circular orthogonal ensemble (COE), the circular unitary
99 ensemble (CUE), or the circular symplectic ensemble (CSE), each gives distinct distributions
100 of energy level spacings

$$P(S) = \begin{cases} \frac{S\pi}{2}\exp\left(-\frac{S^2\pi}{4}\right), & \text{COE}, \\ \frac{S^2 32}{\pi^2}\exp\left(-\frac{S^2 4}{\pi}\right), & \text{CUE}, \\ \frac{S^4 2^{18}}{3^6\pi^3}\exp\left(-\frac{S^2 64}{9\pi}\right), & \text{CSE}, \end{cases} \tag{6}$$

101 corresponding to linear, quadratic, and quartic level repulsion, respectively. Here, $S_n=(E_{n+1}$
102 $-E_n)/\Delta E$ with the $E_n$ being the ordered eigenenergies of $H$ and $\Delta E$ the mean level spacing.
103   Let $\{|b_m\rangle\}_{m=1}^K$ be a basis of the Krylov space formed by eigenstates of the corresponding
104 random matrix associated with the chaotic Hamiltonian $H$. We recall that the Krylov space

105  $\mathcal{K} \equiv \mathrm{span}\{H^n|\psi_0\rangle\}_{n=0}^{\infty}$ is the minimal subspace of the full Hilbert space $\mathcal{H}$ confining the time
106  evolution of $|\psi(t)\rangle_{\mathrm{chaos}} \equiv U(t)|\psi_0\rangle$ where $H \equiv i\ln U(t)/t$ [34]. We write

$$\langle\hat{O}^2(t)\rangle = \sum_{m,m'=1}^{K} a_m^*(t)a_{m'}(t)\langle b_m|\hat{O}^2|b_{m'}\rangle, \qquad (7)$$

107  where $a_m(t) = \langle b_m|\psi_{\mathrm{chaos}}(t)\rangle$, to evaluate Eq. (5). According to the ergodicity hypothesis [32,
108  33], the time averaging for vector $\boldsymbol{a}(t) = [a_1(t), a_2(t), \ldots, a_K(t)]$ as in Eq. (1) can be replaced
109  by averaging over random matrices (that we indicate with an overline in the following). We
110  have $\overline{\langle\hat{O}^2\rangle} = \sum_{m,m'=1}^{K} \overline{a_m^*a_{m'}}\langle b_m|\hat{O}^2|b_{m'}\rangle$ and use the result [62, 64, 65]

$$\overline{a_m^*a_{m'}} = \frac{\Gamma(\beta K/2)}{\Gamma(\beta K/2+1)}\frac{\Gamma(1+\beta/2)}{\Gamma(\beta/2)}\delta_{m,m'}, \qquad (8)$$

111  where $\beta = 1, 2, 4$ refers to COE, CUE, and CSE, respectively, and $\Gamma(\cdot)$ is the Gamma function.
112  Noticing that $\sum_m\langle b_m|\hat{O}^2|b_m\rangle = \mathrm{Tr}_{\mathcal{K}}[\hat{O}^2]$, we find $\overline{\langle\hat{O}^2\rangle} = \mathrm{Tr}_{\mathcal{K}}[\hat{O}^2]/K$, in the limit $K \gg 1$. In the
113  same limit, $\overline{\langle O\rangle^2} = \left(\mathrm{Tr}_{\mathcal{K}}[\hat{O}]\right)^2/K^2 + \mathcal{O}\left(\mathrm{Tr}_{\mathcal{K}}[\hat{O}^2]/K^2\right)$, see Appendix A. Equation (2) is recovered
114  by substituting the above correlation functions into Eq. (5). We emphasize that, in the limit
115  $K \gg 1$, $\beta$-dependent terms do not contribute to the leading terms of QFI (see Appendix A) and
116  Eq. (2) thus universally applies to chaotic dynamics irrespective of whether they are described
117  by COE, CUE, or CSE.

## 3  Chaotic dynamics in the permutation-symmetric subspace

119  We consider $N$-qubit systems with Hamiltonian involving collective spin operators, thus hav-
120  ing $\boldsymbol{J}^2 = \sum_{\alpha=x,y,z} J_\alpha^2$ as a conserved quantity (Casimir invariant). By taking $|\psi_0\rangle$ as a spin-
121  polarized state, we find that the Krylov space is given by the permutation-symmetric subspace
122  $\mathcal{K} = \mathrm{Sym}^N(\mathbb{C}^2)$ of dimension $K = N+1$ with $\boldsymbol{J}^2 = N(N+1)/4\mathbb{I}_{\mathcal{K}}$. In this case,

$$\mathrm{Tr}_{\mathcal{K}}[J_\alpha^2] = N(N+1)(N+2)/12, \quad \mathrm{Tr}_{\mathcal{K}}[J_\alpha] = 0, \qquad (9)$$

123  and thus Eq. (2) predicts $\bar{F}_{\mathrm{chaos}}[J_\alpha] \simeq N^2/3$.

124      We validate these predictions using specific examples of chaotic kicked-top models with
125  all-to-all spin-spin interactions and addressing the different Dyson ensembles. Considering
126  stroboscopic times, we write $|\psi_{\mathrm{chaos}}(n)\rangle = U^n|\psi_0\rangle$, where $U$ is a Floquet operator for the
127  different random matrix models [62, 66]:

$$U_{\mathrm{COE}} = \exp\left(-i\frac{C}{N}J_z^2\right)\exp(-iAJ_x), \qquad (10)$$

$$U_{\mathrm{CUE}} = \exp\left(-i\frac{\lambda'J_y^2}{N}\right)\exp\left(-i\frac{\lambda J_z^2}{N}\right)e^{-ipJ_x}, \qquad (11)$$

$$U_{\mathrm{CSE}} = \exp(-iV)\exp(-iH_0), \qquad (12)$$

128  with $H_0 = 2\lambda_0 J_z^2/N$ and $V = 8\lambda_1 J_z^4/N^3 + 2\lambda_2(J_xJ_z + J_zJ_x)/N + 2\lambda_3(J_xJ_y + J_yJ_x)/N$, for COE,
129  CUE and CSE, respectively.

130      Figures 2 provide direct numerical evidence supporting the $N^2/3$-scaling law for chaotic
131  collective spin dynamics irrespective of the specific random matrix model. Panels (a), (e),
132  and (g) display numerically computed histograms of the level spacing distribution for $i\ln(U_y)$
133  (y=COE, CUE, CSE), and compare them with the analytic expressions Eq. (6). These pan-
134  els clearly distinguish the distinct level-spacing distributions among the three models (COE,

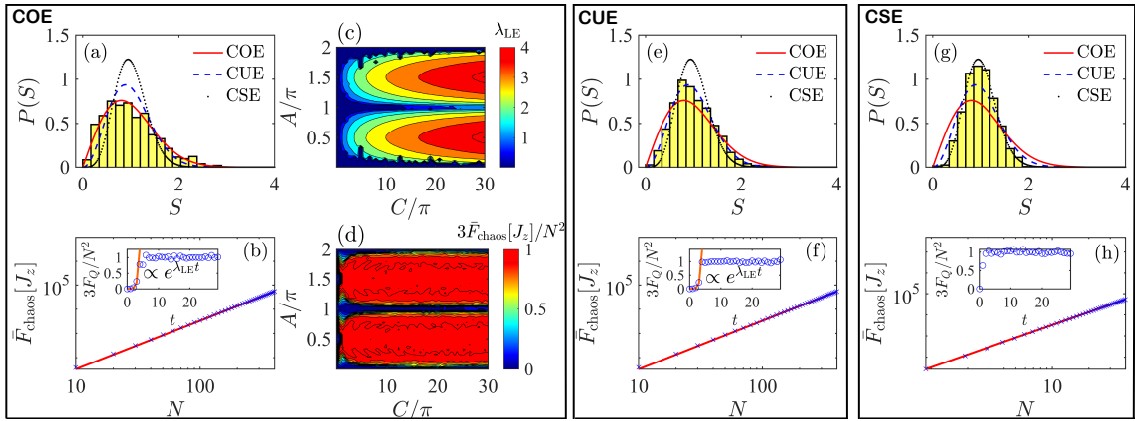

Figure 2: Level spacing distributions $P(S)$ [panels (a), (e), and (g)], LE $\lambda_{\mathrm{LE}}$ [panel (c)], and long-time averaged QFI $\bar{F}_{\mathrm{chaos}}$ [panels (b), (d), (f) and (h)] for COE [panels (a)-(d)], CUE [panels (e)-(f)], and CSE [panels (g)-(h)] chaotic kicked top models (B.1)-(12). Panels (a), (e), and (g) compare numerically obtained level spacing distributions (histograms) for models Eqs. (B.1)-(12) with analytic expressions Eq. (6) [line, dashed line, and dots, respectively], and verify their classification under RMT. Panels (b), (f), and (h) confirm the consistency between numerically obtained $\bar{F}_{\mathrm{chaos}}$ (blue crosses) and the universal scaling $N^2/3$ of QFI from Eq. (2) (red line). Their insets depict the dynamic evolution of QFI (blue circles). The short-time behavior is accurately described by exponential growth (red line) $\propto \exp(\lambda_{\mathrm{LE}} t)$ with $\lambda_{\mathrm{LE}} = 1.874$ for $U_{\mathrm{COE}}$ and $\lambda_{\mathrm{LE}} = 1.937$ for $U_{\mathrm{CUE}}$. After the transient Ehrenfest time, the QFI exhibits barely noticeable fluctuations around the constant value given by Eq. (2). For COE model, the red region in panel (d), given by Eq. (2) corresponds to the region in panel (c) with a positive LE. In all panels, we start with a coherent spin state along the $-y$ direction and evolve it under the following chaotic conditions: $A = 1.7, C = 10$ for model Eq. (B.1), $p = 1.7, \lambda = 10, \lambda' = 0.5$ for model Eq. (B.2), and $\lambda_0 = \lambda_1 = 2.5, \lambda_2 = 5, \lambda_3 = 7.5$ for model Eq. (12).

135  CUE, and CSE). The insets in panels (b), (f), and (h) plot the QFI as a function of time. The
136  QFI $F_Q[|\psi_{\mathrm{chaos}}(t)\rangle, J_z]$ exhibits an initial exponential growth [22, 67] in a short-time regime
137  (for $t \leq t^*$, solid red line) followed by saturation at the predicted value Eq. (2), namely
138  $\bar{F}_{\mathrm{chaos}}[J_\alpha] = N^2/3$. In panels (b), (f), and (h), we plot the QFI extracted from long-time av-
139  eraging for $t \gg t^*$, as a function of the number of qubits $N$. The QFI plotted in all figures
140  corresponds to $J_z$. In the Appendix D, we also plot the QFI with respect to $J_x$ and $J_y$ and
141  numerically verify that the universal scaling of QFI holds for all spin directions. A difference
142  between spin directions is observed only for a short time $t < t^*$, see Appendix D. The numeri-
143  cal results agree very well with Eq. (2) even for relatively small values of $N$. Moreover, panel
144  (c) shows the LE computed numerically, see Appendix B, while panel (d) plots the long-time
145  averaged QFI, as a function of the parameters $A$ and $C$ of the model (B.1). The agreement
146  between the semi-elliptical regions with finite LEs and the region exhibiting $N^2/3$ scaling of
147  the QFI further supports the universality of Eq. (2).

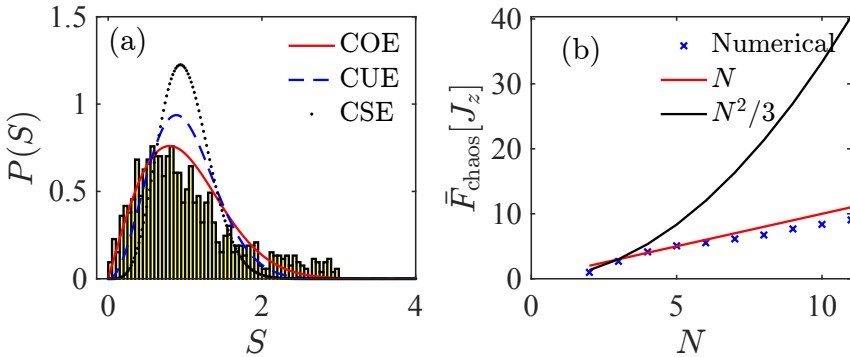

Figure 3: Level spacing distributions [panel (a)] and long-time averaged QFI [panels (b)] for the chaotic Ising model, Eq. (13) with parameters $J = h = \lambda = 1$.

## 4   Chaotic dynamics in the full Hilbert space

Another distinct chaotic model is constructed by applying both transverse and longitudinal fields to the $N$-qubit Ising model

$$H_{\text{cIsing}} = \sum_{i=1}^{N} (J \sigma_i^x \sigma_{i+1}^x + h \sigma_i^x + \lambda \sigma_i^z), \tag{13}$$

where open boundary conditions are adopted. Energy-level spacing statistics, see Fig. 3(a), indicate that this model is chaotic for $J = h = \lambda = 1$ [68, 69]. We argue that the dynamics extend over the Krylov space $\mathcal{K} = \otimes^N \mathbb{C}^2$, which coincides with the full $N$-qubit Hilbert space of dimension $K = 2^N$. In this case, $\text{Tr}_\mathcal{K}[J_\alpha^2] = N 2^{N-2}$, $\text{Tr}_\mathcal{K}[J_\alpha] = 0$, and thus Eq. (2) implies $\bar{F}_{\text{chaos}}[J_\alpha] \simeq N$. This result holds even in the presence of bit reversal symmetry or parity symmetry, which will further reduce the full $N$-qubit Hilbert space, see Appendix A1. Our prediction is confirmed by numerical analysis in Fig. 3(b). More generally, Eq. (2) suggests an operational approach to identify the Krylov space dimension [70] for chaotic dynamics with variable-range interaction, to be explored in future research.

## 5   Comparison with quantum systems with semiclassical correspondence

The QFI of quantum systems with semiclassical correspondence grows exponentially at a rate given by the largest LE, including both chaotic and integrable models [22]. Some integrable models [72, 88] exhibit positive LEs at unstable points, leading to false detection of quantum chaos using OTOCs. However, these models do not conform to the RMT description, indicating they are not truly chaotic and thus invalidating Eq (2). Therefore, compared with OTOCs, our universal scaling Eq. (2), better excludes these atypical integrable models with positive LEs from chaotic models, providing a clearer indicator of quantum chaos.

As an illustration, we examine the Lipkin-Meshkov-Glick (LMG) model [73–75],

$$H_{\text{LMG}} = \Omega J_z - \frac{2\xi}{N} J_x^2, \tag{14}$$

which is characterized by the generalized Gaudin Lie algebra, enabling exact solvability through the Bethe ansatz [76]. It finds experimental realizations in various atomic ensebles [3, 12, 18, 19] and nuclear magnetic resonance platforms [77]. Due to the existence of an unstable point,

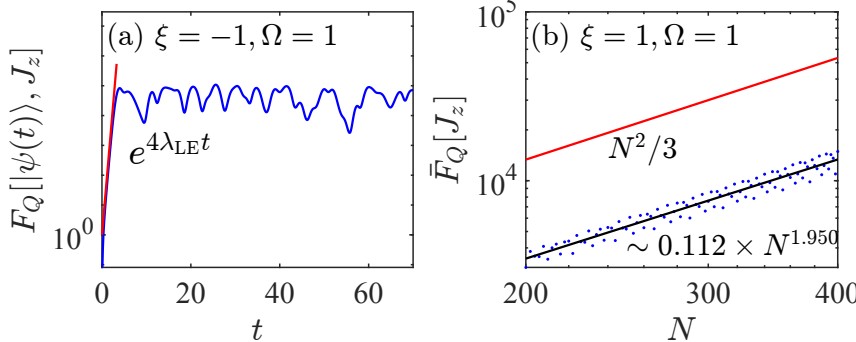

Figure 4: Time-evolution of QFI (a) and long-time averaged QFI (b) in the integrable LMG model for the initial unstable coherent spin state along the $-z$ direction. The red straight line in (a) (semi-logarithmic coordinate) corresponds to the exponential curve with the rate given by four times LE, $\exp(4\lambda_{\text{LE}}t)$. Blue dots in (b) are numerical results, which are fitted by a black solid line with scaling of $F_Q \simeq 0.112 N^{1.950}$. This behavior deviates from the universal scaling for chaotic collective spin models $N^2/3$, as marked by the solid red line.

this integrable model has a positive LE $\lambda_{\text{LE}} = \sqrt{\Omega(2\xi - \Omega)}$ in the phase of $\Omega(\Omega - 2\xi) < 0$ [88], also see Appendix C. Figure 4(a) plots the evolution of the QFI $F_Q[|\psi(t)\rangle, J_\alpha]$ of an initial coherent spin state prepared at the semiclassical unstable fixed point, and for the given $\alpha = z$. Initially, the QFI exhibits exponential growth $\sim e^{4\lambda_{\text{LE}}t}$, resembling chaotic dynamics [22, 36]. However, the saturation value reached for time $t \gg t^*$ does not agree with the value $N^2/3$, Eq.(2), of truly quantum chaotic systems, despite the positive LE. Furthermore, in contrast to Eq. (2) the QFI of the unstable integrable model shows a strong dependence in $\alpha$, see Appendix D.

# 6 Conclusion

Through QFI, we have provided a unifying approach to randomness, multipartite entanglement, and quantum chaos, with a fundamental dependence on the dimension of the Krylov space. We have restricted our discussion to the vast class of chaotic systems governed by Dyson's three ensembles. Nevertheless, we expect possible generalizations beyond the three-fold class [78–82] and open chaotic systems [83], as well as a dependence on possible deviations from RMT statistics. From an application perspective, particularly in quantum metrology, our results highlight the remarkable capability of chaotic collective spin systems to generate scalable multipartite entanglement at an exponentially fast rate. For these systems, the QFI shows a universal scaling $\bar{F}_{\text{chaos}}[J_\alpha] = N^2/3$, indicating the emergence of scalable multipartite entanglement [84]. This observation underscores the potential for achieving Heisenberg-scaling quantum metrology. In contrast to squeezed or GHZ states, which achieve a high QFI only for specific rotation axes [3], Eq. (2) shows that entanglement from chaotic collective spin dynamics is axis-independent and applicable to any initial symmetric separable state, distinguishing it also from entanglement generated by integrable dynamics like axis twisting [9].

## 196  Acknowledgements

197 We would like to thank Satoya Imai, Li Gan, Silvia Pappalardi, Jing Yang, Sandro Wimberger
198 and Wei Xia for discussions.

199 **Funding information**   This work was supported by the European Commission through the
200 H2020 QuantERA ERA-NET Cofund in Quantum Technologies project "MENTA" and received
201 funding under Horizon Europe programme HORIZONCL4-2022-QUANTUM-02-SGA via the
202 project 101113690 (PASQuanS2.1)

## 203  A   Derivation of long-time averaged QFI for chaotic systems, Eq. (2)

204 Let $\{|b_m\rangle\}_{m=1}^{K}$ be a basis of the Krylov space $\mathcal{K} \equiv \text{span}\{H^n|\psi_0\rangle\}_{n=0}^{\infty}$ formed by eigenstates of
205 the corresponding random matrix associated with the chaotic Hamiltonian $H$. The evolved
206 chaotic state $|\psi_{\text{chaos}}(t)\rangle = e^{-iHt}|\psi_0\rangle$ can be expanded in the Krylov space as

$$|\psi_{\text{chaos}}(t)\rangle = \sum_{m=1}^{K} a_m(t)|b_m\rangle, \tag{A.1}$$

207 where $a_m(t) = \langle b_m|\psi_{\text{chaos}}(t)\rangle$. We want to compute

$$\bar{F}_{\text{chaos}}[\hat{O}] \equiv \lim_{T\to\infty} \frac{1}{T-t^*} \int_{t^*}^{T} dt F_Q[|\psi_{\text{chaos}}(t)\rangle, \hat{O}], \tag{A.2}$$

208 where $\hat{O}$ is a Hermitian operator and

$$F_Q[|\psi_{\text{chaos}}(t)\rangle, \hat{O}] = 4\big(\langle\hat{O}^2(t)\rangle - \langle\hat{O}(t)\rangle^2\big) \tag{A.3}$$

209 is the QFI of the evolved chaotic state $|\psi_{\text{chaos}}(t)\rangle$, with

$$\langle\hat{O}^\kappa(t)\rangle = \sum_{m,m'=1}^{K} a_m^*(t)a_{m'}(t)\langle b_m|\hat{O}^\kappa|b_{m'}\rangle. \tag{A.4}$$

210 Combining the above equations we have

$$\begin{aligned}
&\bar{F}_{\text{chaos}}[\hat{O}]\\
&= 4\sum_{m,m'=1}^{K} \langle b_m|\hat{O}^2|b_{m'}\rangle \bigg(\lim_{T\to\infty}\frac{1}{T-t^*}\int_{t^*}^{T} dt\, a_m^*(t)a_{m'}(t)\bigg)\\
&\quad -4\sum_{m_1,m_1',m_2,m_2'=1}^{K} \langle b_{m_1}|\hat{O}|b_{m_1'}\rangle\langle b_{m_2}|\hat{O}|b_{m_2'}\rangle\bigg(\lim_{T\to\infty}\frac{1}{T-t^*}\int_{t^*}^{T} dt\, a_{m_1}^*(t)a_{m_1'}(t)a_{m_2}^*(t)a_{m_2'}(t)\bigg).
\end{aligned} \tag{A.5}$$

211 According to the ergodicity hypothesis [32,33], the time averaging for vector $\boldsymbol{a}(t) = [a_1(t), a_2(t),$
212 $\ldots, a_K(t)]$ can be replaced by averaging over random matrices (that we indicate with an over-
213 line in the following), namely

$$\lim_{T\to\infty} \frac{1}{T-t^*} \int_{t^*}^{T} dt\, a_m^*(t)a_{m'}(t) = \overline{a_m^*(0)a_{m'}(0)},$$

$$\lim_{T\to\infty} \frac{1}{T-t^*} \int_{t^*}^{T} dt\, a_{m_1}^*(t)a_{m_1'}(t)a_{m_2}^*(t)a_{m_2'}(t) = \overline{a_{m_1}^*(0)a_{m_1'}(0)a_{m_2}^*(0)a_{m_2'}(0)}, \tag{A.6}$$

214  where $a_m(0) = \langle b_m | \psi_0 \rangle$ is the projection of the initial state $|\psi_0\rangle$ on the eigenvector $|b_m\rangle$, with
215  $|b_m\rangle$ being the $m$-th eigenvector of one of the random matrices in the random matrix ensemble.
216  We can define another basis formed by $\{|\psi_0\rangle, |\psi_1\rangle, \ldots, |\psi_{K-1}\rangle\}$ and define $a_m \equiv \langle b_1 | \psi_{m-1} \rangle$ as
217  the $m$-th element of the eigenvector $|b_1\rangle$. Random matrix theory (RMT) provides [62, 64, 65]

$$\overline{a_m(0)^* a_{m'}(0)} = \overline{a_m^* a_{m'}} = \overline{|a_m|^2} \, \delta_{m,m'} \tag{A.7}$$

218  and

$$\overline{a_{m_1}^*(0) a_{m_1'}(0) a_{m_2}^*(0) a_{m_2'}(0)} = \overline{a_{m_1}^* a_{m_1'} a_{m_2}^* a_{m_2'}}$$
$$= \overline{|a_m|^4} \, \delta_{m_1,m} \delta_{m_1',m} \delta_{m_2,m} \delta_{m_2',m} + \overline{|a_{m_1}|^2 |a_{m_2}|^2} \left( \delta_{m_1,m_1'} \delta_{m_2,m_2'} + \delta_{m_1,m_2'} \delta_{m_2,m_1'} \right). \tag{A.8}$$

219  where

$$\overline{|a_1|^{2m_1} |a_2|^{2m_2} \cdots |a_K|^{2m_K}} = \frac{\int \left( \prod_{i=1}^K |a_i|^{2m_i} \right) \delta \left( \sum_{i=1}^K |a_i|^2 - 1 \right) \left( \prod_{i=1}^K |a_i|^{\beta-1} \mathrm{d}|a_i| \right)}{\int \delta \left( \sum_{i=1}^K |a_i|^2 - 1 \right) \left( \prod_{i=1}^K \mathrm{d}|a_i| \right)}$$

$$= \frac{\Gamma(\beta K/2)}{\Gamma(\beta K/2 + \sum_{i=1}^K m_i)} \frac{\prod_{i=1}^K \Gamma(m_i + \beta/2)}{\prod_{i=1}^K \Gamma(\beta/2)}. \tag{A.9}$$

220  Here, $m_i$ are non-negative integers, $\Gamma(\cdot)$ is the Gamma function, and $\beta = 1, 2, 4$ refers to COE,
221  CUE and CSE, respectively. Equation A.5 rewrites as

$$\bar{F}_{\text{chaos}}[\hat{O}] = 4(\overline{\langle \hat{O}^2 \rangle} - \overline{\langle \hat{O} \rangle^2}), \tag{A.10}$$

222  with

$$\overline{\langle \hat{O}_\alpha^2 \rangle} = \sum_{m=1}^K \overline{|a_m|^2} \langle b_m | \hat{O}^2 | b_m \rangle, \tag{A.11a}$$

$$\overline{\langle \hat{O} \rangle^2} = \sum_{m=1}^K \overline{|a_m|^4} \langle b_m | \hat{O} | b_m \rangle^2 \tag{A.11b}$$
$$+ \sum_{m_1 \neq m_2}^K \overline{|a_{m_1}|^2 |a_{m_2}|^2} [\langle b_{m_1} | \hat{O} | b_{m_1} \rangle \langle b_{m_2} | \hat{O} | b_{m_2} \rangle + \langle b_{m_1} | \hat{O} | b_{m_2} \rangle \langle b_{m_2} | \hat{O} | b_{m_1} \rangle],$$

223  where the second term is obtained by considering three kinds of nonzeros contributions in
224  Eq. (A.5): $m_1 = m_1' = m_2 = m_2'$, $m_1 = m_1' \neq m_2 = m_2'$, and $m_1 = m_2' \neq m_2 = m_1'$. From Eq. (A.9),
225  we have

$$\overline{|a_m|^2} = \frac{\Gamma(\beta K/2) \Gamma(1 + \beta/2)}{\Gamma(\beta K/2 + 1) \Gamma(\beta/2)} = \frac{1}{K}, \quad \forall m \tag{A.12a}$$

$$\overline{|a_m|^4} = \frac{\Gamma(\beta K/2) \Gamma(2 + \beta/2)}{\Gamma(\beta K/2 + 2) \Gamma(\beta/2)} = \frac{\beta + 2}{(\beta K + 2) K}, \quad \forall m \tag{A.12b}$$

$$\overline{|a_{m_1}|^2 |a_{m_2}|^2} = \frac{\Gamma(\beta K/2) \Gamma(1 + \beta/2) \Gamma(1 + \beta/2)}{\Gamma(\beta K/2 + 2) \Gamma(\beta/2) \Gamma(\beta/2)} = \frac{\beta}{(\beta K + 2) K}, \quad \forall m_1 \neq m_2. \tag{A.12c}$$

226  Substituting Eq. (A.12) into Eq. (A.11), we obtain

$$\overline{\langle \hat{O}^2 \rangle} = \frac{1}{K} \sum_{m=1}^K \langle b_m | \hat{O}^2 | b_m \rangle, \tag{A.13}$$

$$\overline{\langle \hat{O} \rangle^2} = \frac{\beta + 2}{(\beta K + 2) K} \sum_{m=1}^K \langle b_m | \hat{O} | b_m \rangle^2 + \frac{\beta}{(\beta K + 2) K} \sum_{m_1 \neq m_2}^K [\langle b_{m_1} | \hat{O} | b_{m_1} \rangle \langle b_{m_2} | \hat{O} | b_{m_2} \rangle$$
$$+ \langle b_{m_1} | \hat{O} | b_{m_2} \rangle \langle b_{m_2} | \hat{O} | b_{m_1} \rangle], \tag{A.14}$$

227 Due to $\sum_{m=1}^{K} \langle b_m | \hat{O}^2 | b_m \rangle = \text{Tr}_{\mathcal{K}}[\hat{O}^2]$, the first correlation function in Eq. (A.13) becomes

$$\overline{\langle \hat{O}^2 \rangle} \;\; = \;\; \frac{\text{Tr}_{\mathcal{K}}[\hat{O}^2]}{K}, \tag{A.15}$$

228 which is the Eq. (8) in the Main text. The second correlation function in Eq. (A.13) can be
229 evaluated as follows

$$
\begin{aligned}
\overline{\langle \hat{O} \rangle^2} \;\; = \;\; & \frac{\beta}{(\beta K + 2)K} \sum_{m=1}^{K} \langle b_m | \hat{O} | b_m \rangle^2 + \frac{\beta}{(\beta K + 2)K} \sum_{m_1, m_2 = 1}^{K} [\langle b_{m_1} | \hat{O} | b_{m_1} \rangle \langle b_{m_2} | \hat{O} | b_{m_2} \rangle \\
& + \langle b_{m_1} | \hat{O} | b_{m_2} \rangle \langle b_{m_2} | \hat{O} | b_{m_1} \rangle ] \\
= \;\; & \frac{\beta}{(\beta K + 2)K} \sum_{m=1}^{K} \langle b_m | \hat{O} | b_m \rangle^2 + \frac{\beta}{(\beta K + 2)K} \left( \sum_{m=1}^{K} [\langle b_m | \hat{O} | b_m \rangle \right)^2 \\
& + \frac{\beta}{(\beta K + 2)K} \sum_{m_1, m_2 = 1}^{K} \langle b_{m_1} | \hat{O} | b_{m_2} \rangle \langle b_{m_2} | \hat{O} | b_{m_1} \rangle \\
= \;\; & \frac{\beta}{(\beta K + 2)K} \sum_{m=1}^{K} \langle b_m | \hat{O} | b_m \rangle^2 + \frac{\beta \left( \text{Tr}_{\mathcal{K}}[\hat{O}] \right)^2}{(\beta K + 2)K} + \frac{\beta \text{Tr}_{\mathcal{K}}[\hat{O}^2]}{(\beta K + 2)K},
\end{aligned}
\tag{A.16}
$$

230 where in deriving the last equation we have used the identity relation $\sum_{m=1}^{K} | b_m \rangle \langle b_m | = \text{Id}_{\mathcal{K}}$.
231 Let us analyze the three terms in Eq. (A.16) and compare them to Eq. (A.15). Let's start with
232 the first term in Eq. (A.16). We use $\langle b_m | \hat{O} | b_m \rangle^2 \le \langle b_m | \hat{O}^\dagger \hat{O} | b_m \rangle = \langle b_m | \hat{O}^2 | b_m \rangle$, for all $| b_m \rangle$,
233 which follows from the Hermiticity of operator $\hat{O}$. This gives

$$\frac{\beta}{(\beta K + 2)K} \sum_{m=1}^{K} \langle b_m | \hat{O} | b_m \rangle^2 \le \frac{\beta}{(\beta K + 2)K} \sum_{m=1}^{K} \langle b_m | \hat{O}^2 | b_m \rangle = \frac{\beta K}{(\beta K + 2)} \frac{\text{Tr}_{\mathcal{K}}[\hat{O}^2]}{K^2} \le \frac{\text{Tr}_{\mathcal{K}}[\hat{O}^2]}{K^2}. \tag{A.17}$$

234 Similarly, the last term in Eq. (A.16) is

$$\frac{\beta \text{Tr}_{\mathcal{K}}[\hat{O}^2]}{(\beta K + 2)K} = \frac{\beta K}{\beta K + 2} \frac{\text{Tr}_{\mathcal{K}}[\hat{O}^2]}{K^2} \le \frac{\text{Tr}_{\mathcal{K}}[\hat{O}^2]}{K^2}. \tag{A.18}$$

235 Finally, we will show that the second term in Eq. (A.16) is the same or lesser order term than
236 $\text{Tr}_{\mathcal{K}}[\hat{O}^2]/K$, and thus it should be kept. By writing the operator $\hat{O}$ in the spectral decomposition
237 form within the Krylov space: $\hat{O} = \sum_{m=1}^{K} \lambda_m | d_m \rangle \langle d_m |$. It follows that

$$\text{Tr}_{\mathcal{K}}[\hat{O}^2] = \sum_{m=1}^{K} \lambda_m^2, \quad (\text{Tr}_{\mathcal{K}}[\hat{O}])^2 = \left( \sum_{m=1}^{K} \lambda_m \right)^2. \tag{A.19}$$

238 Applying Jensen's inequality for the convex function $f(x) = x^2$, we obtain

$$\frac{1}{K} \sum_{m=1}^{K} \lambda_m^2 \ge \left( \frac{1}{K} \sum_{m=1}^{K} \lambda_m \right)^2 \tag{A.20}$$

239 Combining Eqs. (A.19) and (A.20), we deduce that

$$K \text{Tr}_{\mathcal{K}}[\hat{O}^2] \ge (\text{Tr}_{\mathcal{K}}[\hat{O}])^2. \tag{A.21}$$

240 Thus, the second term in Eq. (A.16) is

$$\frac{\beta \left( \text{Tr}_{\mathcal{K}}[\hat{O}] \right)^2}{(\beta K + 2)K} \le \frac{\beta}{\beta K + 2} \text{Tr}_{\mathcal{K}}[\hat{O}^2] \le \frac{\text{Tr}_{\mathcal{K}}[\hat{O}^2]}{K}, \tag{A.22}$$

which implies that this term $\beta \left(\text{Tr}_{\mathcal{K}}[\hat{O}]\right)^2/(\beta K + 2)K$ is the same or lesser order term than $\text{Tr}_{\mathcal{K}}[\hat{O}^2]/K$. The following examples will show that $\beta \left(\text{Tr}_{\mathcal{K}}[\hat{O}]\right)^2/(\beta K + 2)K$ can be on the same order of $\text{Tr}_{\mathcal{K}}[\hat{O}^2]/K$ and thus it will contribute to the leading terms of QFI for some special situations. Based on the above analysis, Eq. (A.16) can be written as

$$\overline{\langle \hat{O} \rangle^2} = \frac{\beta \left(\text{Tr}_{\mathcal{K}}[\hat{O}]\right)^2}{(\beta K + 2)K} + \mathcal{O}\left(\frac{\text{Tr}_{\mathcal{K}}[\hat{O}^2]}{K^2}\right). \tag{A.23}$$

To analyze the influence of $\beta$, we can expand $\beta/(\beta K + 2)K$ by using Taylor expansion up to $\beta$, i.e.,

$$\frac{\beta}{K(\beta K + 2)} = \frac{1}{K^2}\left(1 + \frac{2}{\beta K}\right)^{-1} = \frac{1}{K^2} - \frac{2}{\beta K^3} + \mathcal{O}(\frac{1}{K^4}). \tag{A.24}$$

Therefore, Eq. (A.23) becomes

$$\begin{aligned}
\overline{\langle \hat{O} \rangle^2} &= \frac{\left(\text{Tr}_{\mathcal{K}}[\hat{O}]\right)^2}{K^2} - \frac{2\left(\text{Tr}_{\mathcal{K}}[\hat{O}]\right)^2}{\beta K^3} + \mathcal{O}\left(\frac{\text{Tr}_{\mathcal{K}}[\hat{O}^2]}{K^2}\right) \\
&= \frac{\left(\text{Tr}_{\mathcal{K}}[\hat{O}]\right)^2}{K^2} + \mathcal{O}\left(\frac{\text{Tr}_{\mathcal{K}}[\hat{O}^2]}{K^2}\right),
\end{aligned} \tag{A.25}$$

where the $\beta$-dependent term $\left(\text{Tr}_{\mathcal{K}}[\hat{O}]\right)^2/\beta K^3 \leq \text{Tr}_{\mathcal{K}}[\hat{O}^2]/\beta K^2$ cannot contribute to the leading terms.

Combining Eqs. (A.10), (A.15) and (A.25), we find

$$\bar{F}_{\text{chaos}}[\hat{O}] = \frac{4\text{Tr}_{\mathcal{K}}[\hat{O}^2]}{K} - \frac{4\left(\text{Tr}_{\mathcal{K}}[\hat{O}]\right)^2}{K^2} + \mathcal{O}\left(\frac{\text{Tr}_{\mathcal{K}}[\hat{O}^2]}{K^2}\right), \tag{A.26}$$

which is one of the main results of our manuscript, Eq. (2) in the main text. Next, we will discuss different behaviors of QFI by considering two kinds of Krylov spaces.

## A.1   Full Hilbert space with or without bit reversal symmetry or parity symmetry

For a fix collective spin operator $J_\alpha$, we can define a basis $\{|j_1, j_2, \cdots, j_N\rangle\}$, $(j_\ell \in \{0,1\}, \ell = 1, 2, \ldots, N)$ when considering the full Hilbert space $\otimes^N \mathbb{C}^2$ as the Krylov space, denoted as $\mathcal{K}_f$. Clearly, the dimension of $\mathcal{K}_f$ is $K = 2^N$. By noticing that $|j_1, j_2, \cdots, j_N\rangle$ is an eigenvector of $J_\alpha$ with an eigenvalue $\sum_{\ell=1}^{N}(2j_\ell - 1)/2$, then we have

$$\text{Tr}_{\mathcal{K}_f}[J_\alpha] = \sum_{m=0}^{N}\left(m - \frac{N}{2}\right)\binom{N}{m} = 0, \ \text{Tr}_{\mathcal{K}_f}[J_\alpha^2] = \sum_{m=0}^{N}\left(m - \frac{N}{2}\right)^2\binom{N}{m} = 2^N\frac{N}{4}, \tag{A.27}$$

where we have used the results

$$\sum_{m=0}^{N}\binom{N}{m} = 2^N, \quad \sum_{m=0}^{N} m\binom{N}{m} = 2^N\frac{N}{2}, \quad \sum_{m=0}^{N} m^2\binom{N}{m} = 2^N\frac{N(N+1)}{4}. \tag{A.28}$$

Therefore, by Eq. (A.26), we find $\bar{F}_{\text{chaos}}[J_\alpha] = N + \mathcal{O}(N/2^N)$ for the full Hilbert space.

In the following we want to show that $\bar{F}_{\text{chaos}}[J_\alpha] \simeq N$ for the full Hilbert space with or without bit reversal symmetry or parity symmetry. Let us first include parity symmetry. This separates the full Hilbert space into two subspaces constructed by states with an odd number

of up spins or an even number of up spins. For the parity-odd (parity-even) subspace as the Krylov space, denoted as $\mathcal{K}_+$ ($\mathcal{K}_-$), the basis will be $\{|j_1, j_2, \cdots, j_K\rangle\}$ with $\sum_{\ell=1}^{N} j_\ell$ being odd (even). Therefore, we have

$$\text{Tr}_{\mathcal{K}_\pm}[\boldsymbol{J}_\alpha] = \sum_{m=0}^{\lfloor N/2 \rfloor} \left( 2m + \frac{1 \pm 1}{2} - \frac{N}{2} \right) \binom{N}{2m} = 2^N \frac{1 \pm 1}{4},$$

$$\text{Tr}_{\mathcal{K}_\pm}[\boldsymbol{J}_\alpha^2] = \sum_{m=0}^{\lfloor N/2 \rfloor} \left( 2m + \frac{1 \pm 1}{2} - \frac{N}{2} \right)^2 \binom{N}{2m} = 2^N \frac{N + (1 \pm 1)^2}{8}, \quad \text{(A.29)}$$

where we have used the results

$$\sum_{m=0}^{\lfloor N/2 \rfloor} \binom{N}{2m} = 2^N \frac{1}{2}, \quad \sum_{m=0}^{\lfloor N/2 \rfloor} m \binom{N}{2m} = 2^N \frac{N}{8}, \quad \sum_{m=0}^{\lfloor N/2 \rfloor} m^2 \binom{N}{2m} = 2^N \frac{N(N+1)}{32}, \ \forall \ N \geq 3. \ \text{(A.30)}$$

The dimensions of $\mathcal{K}_\pm$ are both $2^{N-1}$. Therefore, substituting Eq. (A.29) into Eq. (A.26), we conclude that $\bar{F}_{\text{chaos}}[\boldsymbol{J}_\alpha] = N + \mathcal{O}(N/2^N)$ for both spaces $\mathcal{K}_\pm$.

Finally, we consider a full Hilbert space subjected to the bit reversal symmetry $B$. The bit reversal operator $B$ is defined by

$$B|j_1, j_2, \ldots, j_N\rangle = |j_N, j_{N-1}, \ldots, j_1\rangle. \quad \text{(A.31)}$$

According to the bit reversal symmetry, the separated subspaces are denoted by $\mathcal{K}_{b\pm}$ corresponding to $B = \pm 1$. For simplification, we only consider the case of $N$ being even. The dimension of $\mathcal{K}_{b-}$ is half the number of nonpalindromic binary words of length $N$, i.e., $2^{N-1} - 2^{N/2-1}$, and the dimension of $\mathcal{K}_{b+}$ is thus $2^{N-1} + 2^{N/2-1}$. Introducing the space $\mathcal{P} \equiv \text{span}\{|j_1, j_2, \cdots, j_N\rangle :$ $B|j_1, j_2, \cdots, j_N\rangle = |j_1, j_2, \cdots, j_N\rangle\}$. It is straightforward to prove that

$$\text{Tr}_{\mathcal{K}_{b+}}[\boldsymbol{J}_\alpha^\eta] = \text{Tr}_{\mathcal{P}}[\boldsymbol{J}_\alpha^\eta] + \frac{\text{Tr}_{\mathcal{K}_f}[\boldsymbol{J}_\alpha^\eta] - \text{Tr}_{\mathcal{P}}[\boldsymbol{J}_\alpha^\eta]}{2}, \quad \text{Tr}_{\mathcal{K}_{b-}}[\boldsymbol{J}_\alpha^\eta] = \frac{\text{Tr}_{\mathcal{K}_f}[\boldsymbol{J}_\alpha^\eta] - \text{Tr}_{\mathcal{P}}[\boldsymbol{J}_\alpha^\eta]}{2}, \quad \text{(A.32)}$$

where $\eta = 1, 2$. An direction calculation gives

$$\text{Tr}_{\mathcal{P}}[\boldsymbol{J}_\alpha] = \sum_{m=0}^{N/2} \left( 2m - \frac{N}{2} \right) \binom{N/2}{m} = 0, \ \text{Tr}_{\mathcal{P}}[\boldsymbol{J}_\alpha^2] = \sum_{m=0}^{N/2} \left( 2m - \frac{N}{2} \right)^2 \binom{N/2}{m} = 2^{N/2} \frac{N}{2}. \ \text{(A.33)}$$

Substituting Eqs. (A.33) and (A.27) into Eq. (A.32) we have

$$\text{Tr}_{\mathcal{K}_{b\pm}}[\boldsymbol{J}_\alpha] = 0, \quad \text{Tr}_{\mathcal{K}_{b\pm}}[\boldsymbol{J}_\alpha^2] = 2^N \frac{N}{8} \pm 2^{N/2} \frac{N}{4}. \quad \text{(A.34)}$$

Therefore, further by Eq. (A.26) we conclude that $\bar{F}_{\text{chaos}}[\boldsymbol{J}_\alpha] = N + \mathcal{O}(N/2^N)$ for both spaces $\mathcal{K}_{b\pm}$.

## A.2 Permutation-symmetric subspace

Now we consider the permutation-symmetric subspace as the Krylov space, denoted by $\mathcal{K}_s$. For a fixed spin operator $\boldsymbol{J}_\alpha$, we can construct a basis for $\mathcal{K}_s$ by using the Dickes state $\{|m\rangle\}_{m=-N/2}^{N/2}$ and meanwhile ensure $\boldsymbol{J}_\alpha|m\rangle = m|m\rangle$. It finds that the dimension of $\mathcal{K}_s$ is $N + 1$ and

$$\text{Tr}_{\mathcal{K}_s}[\boldsymbol{J}_\alpha] = \sum_{m=-N/2}^{N/2} m = 0, \quad \text{Tr}_{\mathcal{K}_s}[\boldsymbol{J}_\alpha^2] = \sum_{m=-N/2}^{N/2} m^2 = \frac{N(N+1)(N+2)}{12}. \quad \text{(A.35)}$$

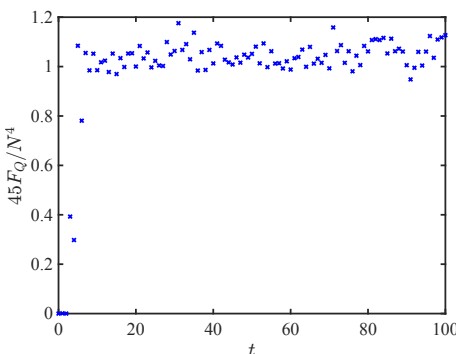

Figure 5: Evolution of QFI with respect to $J_z^2$ in the chaotic COE model with parameters $A = 1.7$ and $C = 10$.

Then by using Eq.(A.26), we find that $\bar{F}_{\text{chaos}}[\boldsymbol{J}_\alpha] = N^2/3 + \mathcal{O}(N)$.

Furthermore, we can consider the case of taking $\hat{O} = \boldsymbol{J}_\alpha^2$. Due to

$$\text{Tr}_{\mathcal{K}_s}[\boldsymbol{J}_\alpha^4] = \sum_{m=-N/2}^{N/2} m^4 = \frac{N(N+1)(N+2)(3N^2 + 6N - 4)}{240}, \tag{A.36}$$

then by Eq. (A.26), we have

$$\bar{F}_{\text{chaos}}[\boldsymbol{J}_\alpha^2] = \frac{N^4}{45} + \mathcal{O}(N^3), \tag{A.37}$$

which is numerically verified in Fig. 5.

# B  Calculation of Lyapunov Exponent (LE)

The chaotic properties of quantum chaotic models are quantitatively assessed by their (largest) LE, denoted as $\lambda_{\text{LE}}$, from their classical counterpart, which measures the rate of divergence between nearby orbits in classical dynamical systems. We take the COE-described kicked top model as an example to illustrate the numerical calculation of LE. Its Hamiltonian is given by

$$U_{\text{COE}} = \exp\left(-i\frac{C}{N}J_z^2\right)\exp(-iAJ_x), \tag{B.1}$$

$$\tag{B.2}$$

by defining classical variable $\vec{X} \equiv (X, Y, Z) = 2(J_x, J_y, J_z)/N$, quantum dynamics $\vec{J}_{n+1} = U_{\text{COE}}^\dagger \vec{J}_n U_{\text{COE}}$ can be mapped onto classical dynamics in the limit $N \to \infty$, as follows

$$\begin{aligned}
X_{n+1} &= X_n \cos\Theta_n - (\cos AY_n - \sin AZ_n)\sin\Theta_n, \\
Y_{n+1} &= X_n \sin\Theta_n + (\cos AY_n - \sin AZ_n)\cos\Theta_n, \\
Z_{n+1} &= \cos AZ_n + \sin AY_n,
\end{aligned} \tag{B.3}$$

where $\Theta_n = C(\cos AZ_n + \sin AY_n)$. The LE can be calculated by using [85–87]

$$\lambda_{\text{LE}} = \ln\left[\lim_{n\to\infty}(t_+(n))^{1/n}\right], \tag{B.4}$$

where $t_+(n)$ is the largest eigenvalue of $T = \prod_{p=1}^{n}\mathcal{T}(\vec{X}_p)$ and $\mathcal{T}(\vec{X}_n) = \partial\vec{X}_{n+1}/\partial\vec{X}_n$. In the main text, we will always consider the following coherent spin state as the initial state for

numerical calculation,

$$
\begin{aligned}
|\theta, \phi\rangle &= \cos^N\left(\frac{\theta}{2}\right)\exp\left[\tan\left(\frac{\theta}{2}\right)e^{i\phi}J^-\right]|N/2, N/2\rangle \\
&= \bigotimes_{i=1}^{N}\left(\cos\frac{\theta}{2}|\uparrow\rangle_i + e^{i\phi}\sin\frac{\theta}{2}|\downarrow\rangle_i\right).
\end{aligned}
\tag{B.5}
$$

Therefore, the initial classical variables for coherent spin states are

$$
X_0 = \sin(\theta)\cos(\phi),\ Y_0 = \sin(\theta)\sin(\phi),\ Z_0 = \cos(\theta).
\tag{B.6}
$$

Equations (B.3) and (B.4) provide a numerical approach to calculate LE.

## C  An unstable point in the LMG model

The Lipkin-Meshkov-Glick (LMG) model is given by

$$
H_{\mathrm{LMG}} = \Omega J_z - \frac{2\xi}{N}J_x^2.
\tag{C.1}
$$

To identify unstable points, we express the LMG Hamiltonian with respect to coherent spin states

$$
\begin{aligned}
h_{\mathrm{LMG}} &\equiv \frac{\langle\theta, \phi|H_{\mathrm{LMG}}|\theta, \phi\rangle}{N/2} \\
&= \Omega\cos\theta - 2\xi\sin^2\theta\cos^2\phi.
\end{aligned}
\tag{C.2}
$$

Canonical variables are defined as follows $Q = \sqrt{2(1+\cos\theta)}\cos\phi$ and $P = -\sqrt{2(1+\cos\theta)}\sin\phi$. Consequently, Eq. (C.2) transforms into

$$
h_{\mathrm{LMG}} = \frac{\Omega}{2}(P^2 + Q^2) - \Omega - \xi Q^2\left(1 - \frac{P^2 + Q^2}{4}\right).
\tag{C.3}
$$

Analysis of the classical Hamiltonian, Eq. (C.2), reveals that $P = Q = 0$ denotes the unstable point when $\Omega(\Omega - 2\xi) < 0$ since $\partial_P h_{\mathrm{LMG}}(0,0) = \partial_Q h_{\mathrm{LMG}}(0,0) = 0$ and $\partial_P^2 h_{\mathrm{LMG}}(0,0)$ $\partial_Q^2 h_{\mathrm{LMG}}(0,0) - [\partial_P\partial_Q h_{\mathrm{LMG}}(0,0)]^2 = \Omega(\Omega - 2\xi) < 0$. Correspondingly, the positive Lyapunov Exponent (LE) for this unstable point, $P = Q = 0$, is given by [88]

$$
\lambda_{\mathrm{LE}} = \sqrt{\Omega(2\xi - \Omega)}.
\tag{C.4}
$$

## D  QFI for different spin directions

From Fig. (6), we see that in chaotic collective spin models, e.g., $U_{\mathrm{COE}}$, $U_{\mathrm{CUE}}$, and $U_{\mathrm{CSE}}$, defined in the main text, the short-time behavior of QFI is dependent on the choice of the direction of the spin operator $J_\alpha$. However, in the long-time region, QFI in chaotic collective models behaves independently of $\alpha$ and tends towards a universal behavior, $F_Q \to N^2/3$. For the LMG model, although at the unstable point, QFI displays exponential growth during the short time, there is no universal behavior of QFI even in the long-time region, and it highly depends on the choice of $\alpha$.

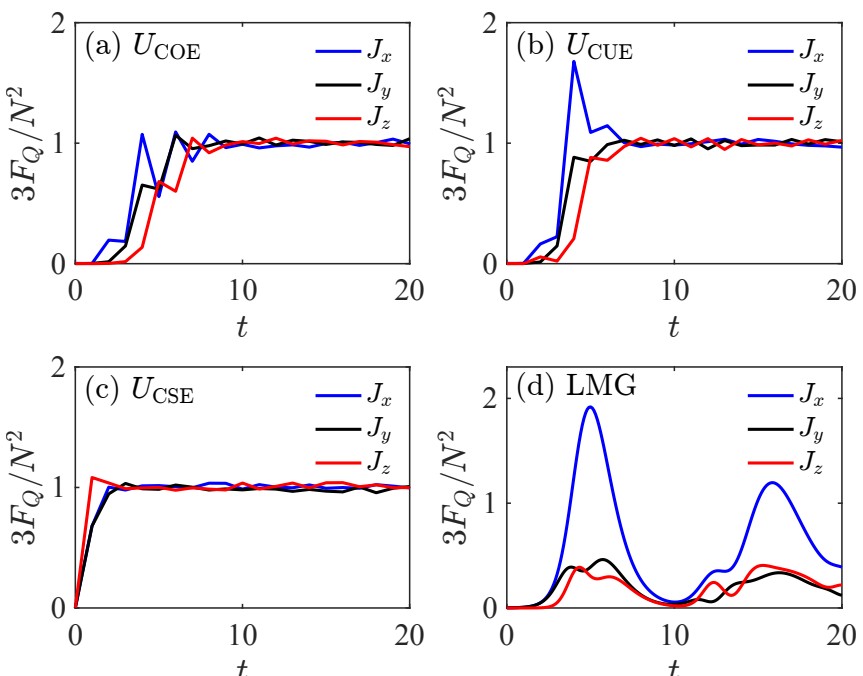

Figure 6: Evolution of QFI with respect to different spin operators $J_{x,y,z}$ in models of (a) $U_{\mathrm{COE}}$, (b) $U_{\mathrm{CUE}}$, (c) $U_{\mathrm{CSE}}$, and (d) LMG. Chaotic conditions are used, $A = 1.7, C = 10$ for (a), $p = 1.7$, $\lambda = 10$, $\lambda' = 0.5$ for (b), and $\lambda_0 = \lambda_1 = 2.5$, $\lambda_2 = 5$, and $\lambda_3 = 7.5$ for (c). The coherent spin state $|\pi/2, -\pi/2\rangle$ is used in panels (a-c). In panel (d), we consider the unstable coherent spin state $|\pi, \phi\rangle$ as the initial state, and parameters of the LMG model are chosen as $\Omega = 1$ and $\xi = -1$.

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
