# Peer review of "Quantum Chaos, Randomness and Universal Scaling of Entanglement in Various Krylov Spaces"

_SciPost Physics, doi:SciPost Phys. 19, 102 (2025)_

## Round 1 · Referee Report · Anonymous (Referee 1) · 2025-2-19

Strengths

  1. Extend the observation, that random states may have large QFI from [38] to a chaotic evolution case.
  2. One of the main results Eq. (2) is very general, applicable for the broad class of models.
  3. The general theorem is nicely supported with analysis of the examples with useful numerical plots (easily readable and didactic).

Weaknesses

  1. Unclear relation to the references [26-28].
  2. Unreliable discussion on application in quantum metrology in Conclusion.

Report

The report is attached as a pdf file.

Requested changes

The requested changes are listed in the report pdf file.

Attachment

Recommendation

Ask for minor revision

---

## Round 1 · Referee Report · Anonymous (Referee 2) · 2025-3-3

Strengths

  • Very clear presentation.
  • Clear result.
  • New knowledge on use of various chaos indicators.
  • Possible relevance for experiments.

Weaknesses

The various chaos measures could be differentiated better.

Report

The message of this paper is very clear and the presentation is excellent. I suggest publication in this open access journal after some minor suggestions have been addressed which I list in the following: 1) very few formal problems should be solved, e.g. ref. 88 is already cited as ref. 71, and Fig. 2 I suggest to divide into two subplots for better readability. 2) what I learn from this paper is that OTOCS do predict the correct short-term evolution (LE etc.) but may not be the best indicator for long-term mixing. This asymptotic statement might be differentiated a bit better together with other possible differences between the various chaos measures.

Requested changes

See report 1)-2).

Recommendation

Publish (easily meets expectations and criteria for this Journal; among top 50%)

---

## Round 2 · Author Response

The message of this paper is very clear and the presentation is excellent. I suggest publication in this open access journal after some minor suggestions have been addressed which I list in the following: 1) very few formal problems should be solved, e.g. ref. 88 is already cited as ref. 71, and Fig. 2 I suggest to divide into two subplots for better readability. 2) what I learn from this paper is that OTOCS do predict the correct short-term evolution (LE etc.) but may not be the best indicator for long-term mixing. This asymptotic statement might be differentiated a bit better together with other possible differences between the various chaos measures.
Reply: We thank the Referee for the careful reading and the positive assessment of our work. We also appreciate the Referee's comments regarding the referencing issues. These problems have been addressed in the revised version. Figure 2 has been revised to improve readability. As the Referee pointed out, distinguishing fast-scrambling integrable dynamics from chaotic dynamics based on short-time correlation functions-such as OTOCs-can be challenging. In contrast, QFI provides a long-time diagnostic and successfully differentiates between them. Accordingly, we now emphasize this point at the end of the Conclusion.
Report of Referee B: The Authors analyzed time-averaged quantum Fisher information (which can be a testimony of multipartite entanglement) for the chaotic models described by Dyson's three circular ensembles. They derive the simple and general asymptotic formula for averaged QFI Eq. (2) (by `asymptotic' I mean valid for large dimensions of Krylov space), in line with intuition and expectations. Next, using both analytical argument and numerical simulations, they perform a detailed analysis of symmetric N-qubit space for three ensembles taken from ref. [62], showing Heisenberg scaling of QFI (which is indeed a testimony of multipartite entanglement). Finally, they compared the results with different chaotic models and the model with semiclassical correspondence, showing, that the previous statements cannot be extended for the latter models. The fact, that the random state can have large QFI has been noticed and broadly explored in 2016 in [38], where Haar-random states have been analyzed. The Authors extend it to the case of chaotic dynamics in a correct and methodical way. Therefore, I would say that the paper ``provides a link between different research areas'' and ``opens a new pathway in an existing direction, with clear potential for multi-pronged follow-up work'', so it satisfies Scipost criteria. However, I do have some reservations about the presentation of the results and some of the relationships with the literature that need to be addressed before I recommend publication.
Reply: We thank the Referee for the careful reading and the positive assessment of our work. Below is our point-by-point response to Referee B's questions.
Report of Referee B: 1. (optional, but strongly recommended) One of the main results of the paper is formula Eq. (2), which is meaningful only in the limit of large $K$. It would be profitable to derive and present the exact lower and upper bound for QFI with the exact constant multiplying the term ${\rm Tr}[O^2]/K^2$, instead of saying, that it is equality up to the term of order ${\rm Tr}[O^2]/K^2$. Looking at App.A, it seems that it does not require much additional work, but it would be extremely helpful for the reader to judge, for which $K$ the formula is already a reasonable approximation.
Reply: We thank the Referee for the insightful comment. However, we note that our Eq.(2) already includes the two leading contributions, while the final subleading term is beyond the predictive power of random matrix theory (RMT), for the following reason. In quantum chaotic many-body systems, collective spin operators $J_\alpha$ satisfy the commutation relation $[J_\alpha/N,J_\beta/N]=i\epsilon_{\alpha\beta\gamma}J_\gamma/N$, which vanishes in the thermodynamic limit $N\to\infty$. This asymptotic commutativity implies that normalized observables $J_\alpha/N$ behave effectively as classical variables in this limit. Consequently, second-order correlations such as $\langle J_\alpha J_\beta\rangle/N^2$ are well described by RMT, which captures the typical fluctuation behavior of chaotic systems under the appropriate symmetry class. In contrast, subleading terms involving correlations, such as $\langle J_\alpha J_\beta\rangle/N^3$, encode finer details of the dynamics and generally fall beyond the predictive capacity of RMT. For clarity, we focus on the case $K=N$, where the final term in Eq.~(2), scaling as $\mathcal O({\rm Tr}[O^2]/K^2)$, lies outside the RMT regime. Our theoretical predictions, therefore, include all leading-order contributions accessible via RMT, and we find that they already yield excellent agreement with numerical simulations (see Fig.~3), even for relatively small $K=2^N$, $N=2,\ldots, 11$. In the revised version, we have emphasized this point more clearly.
Report of Referee B: 2. In Figure 2. the Authors present the perfect correspondence of the numerical results and analytical predictions for the large $N$. This is highly expected due to applied theorems. It would be profitable to show also the results for smaller $N$ to see, how the asymptotic formula Eq. (2) becomes valid (i.e., what is the smallest $N$, to make it reasonable). That would also allow us to see the differences in averaged QFI for different ensembles (which disappear with increasing $N$).
Reply: We recall that Fig.~2 refers to states in the permutation-symmetric subspace. Although, for $N$ qubits, the Hilbert space dimension is nominally large, $2^{N}$, due to the permutation symmetry present in the system, the effective dimension of the Krylov subspace is significantly reduced to approximately $K=N+1$. Therefore, even with a relatively small Krylov subspace dimension $K$, our results still show good agreement and predictive power. To clarify this, we have extended the plot from $N=2$.
Report of Referee B: 3. In the introduction the Authors refer to [26-28], ending the paragraph with the conclusion ``It thus remains unclear which types of chaotic systems can generate scalable multipartite entanglement, i.e., QFI scaling as $N^2$ known as the Heisenberg scaling.''. All references [26-28] refer to the optimal metrology strategy where both the number of particles $N$ , but also total time $t$ are treated as resources. For example, in [27] we read ``In this scenario, even if the system might be able to be prepared in a global multipartite entangled state with $F_Q\sim N^2$, the Heisenberg scaling would not be feasible due to the required state preparation time that scales at least linearly with the system size''. Therefore, [27] does not exclude the possibility of obtaining $F_Q\sim N^2$; that only said, that it would be not useful for metrology (as it takes a long time, which could be more efficiently spent for performing more repetitions of experiments). Do the Authors claim, that their results stay in consistent, contradict, or are completely unrelated to the ones from [26-28]? Please clarify this relation.
Reply: We thank the Referee for the thoughtful and careful comment. In Ref. [27], the author employed the Lieb-Robinson bound to estimate the minimum evolution time required to prepare multipartite entangled states exhibiting Heisenberg scaling (HS) from initially separable states, under the assumption that such HS-target states exist in the system. In Ref. [26], it was rigorously shown that when estimating a magnetic field using short-range local Hamiltonians and separable initial states, the final QFI scales as $F_Q\sim N$. This implies that not all many-body systems can dynamically generate HS states from separable states, even with long-time evolution. Ref. [28] focused on the rapid generation of multipartite entanglement from separable initial states, emphasizing the role of correlation spreading-closely related to the concept of fast scrambling-as a necessary mechanism. In our work, we derive the long-time behavior of QFI in chaotic systems and find that symmetry plays a crucial role in enabling the emergence of HS scaling. For instance, although fast scrambling occurs in the chaotic Ising model, we observe that HS scaling cannot be achieved from initially separable states due to the lack of the relevant symmetry structure. At the end of the Conclusion and Discussion section, we have added a new paragraph to clarify the connections between our work and the related literature.
Report of Referee B: 4. Regarding the `Conclusion' section, I really doubt the potential for application in metrology and I find the comparison with GHZ states extremely dishonest. In the paper, the Author's analyze the averaged value of QFI calculated for pure stated obtainable by chaotic evolution. To use such a state in metrological tasks, we need to have full control of the chaotic Hamiltonian and to know perfectly the time of evolution chaotic evolution, because optimal measurement and the estimator may depend on the exact state. Prepering the states by performing chaotic evolution seems to be an extremely unstable method. So even if it could be possible to perform proof-of-principle experiments, I cannot find an example, when using this theorem would be useful for real metrological problems.
}
Replay: We respectfully disagree with the Referee: we believe that chaotic states prepared in the permutation-symmetric subspace have strong potential for real metrological problems. First, our approach does not require precise knowledge of the evolution time: it only requires that the system evolves beyond the Ehrenfest time to ensure that the resulting state is chaotic. Second, while GHZ states can in principle achieve Heisenberg-limited precision in phase estimation, their preparation demands full control over the many-body Hamiltonian to engineer specific superpositions such as $|0\cdots 0\rangle+|11\cdots 1\rangle$. Moreover, the QFI of a GHZ state reaches $N^2$ only for phase shifts generated by $J_z$, but vanishes for $J_x$ or other orthogonal directions. Therefore, high-precision estimation is only possible along a specific axis. In contrast, chaotic states exhibit a QFI of $N^2/3$ uniformly for any spin direction, allowing for axis-independent phase estimation. This directional flexibility not only enhances robustness but also eliminates the need for precise time control or fine-tuned Hamiltonians-one only needs to ensure that the system evolves into a chaotic state.
Report of Referee B: 5. Next, regarding GHZ states-it is of course true that any single GHZ states achieve high QFI for only one specific axis. The authors claim, that using chaotic systems gives an advantage, as the averaged QFI scales like $N^2/3$ for all axes. But this is an unfair comparison - if one considers the set of three different GHZ states, for them the average QFI will be also $N^2/3$. It would be worth stressing, that what the Authors analyzed is the average value of QFI for pure states, not the value of QFI for the averaged (mixed) stated. Therefore, an operational understanding of their results corresponds to the situation, where different measurements are performed at different times, with the perfect knowledge of the time of performing measurement. It is not that they have found a single state useful for measuring rotation for all axes - they only show that during chaotic evolution different states are good for measuring rotation around different axes, so on average QFI for all axes is large. But exactly the same effect may be obtained for the usage of different GHZ states. So where is the advantage? (optional) If the Authors are interested in showing the true superiority of random chaotic states over the GHZ states, I recommend analyzing for example noise resistance (as done in [38] for random Haar state).
Reply: We respectfully disagree. The Referee suggests to consider the set of three different GHZ states: however every GHZ state of the set will be useful only along a specific axis. Overall, the set will be useful along three axes, while our chaotic states are useful around {\it any} axes. The Referee's concern likely arises from a misunderstanding related to the time-averaged nature of the QFI. We emphasize that, for chaotic dynamics, the QFI at times beyond the Ehrenfest time closely approaches its long-time average value, as shown in Fig. 6(a-c). In this sense, a single evolved chaotic state suffices for phase estimation with high precision in any spin direction, yielding $\Delta\theta\sim \sqrt{3}/N$ corresponding to $F_Q=N^2/3$. Compared to the GHZ state, this represents only a modest reduction in precision by a constant prefactor. However, the key advantage is that no specific measurement direction is required, which significantly simplifies experimental implementation.
%While it is possible to construct an axis-independent QFI of using three orthogonal GHZ states, preparing such a set demands fine-tuned control over the Hamiltonian and coherent access to multiple entangled states, posing a considerable experimental challenge.
Report of Referee B: 6. Besides that, I have so minor or technical comments:(A) Before Eq. (4), the definition of fidelity is written for the very strange form to me. I.e., it trivially simplifies to $\langle\psi(\theta)|W^\dag(\theta,t)|\psi(\theta)\rangle$, or even simpler $\langle\psi(\theta)|W^\dag(\theta,t)|\psi(\theta)\rangle$ (which is the most common form for pure states). Is the usage of the longer and more complicated form is intentional? (B) The Figure 2. is hard to read after printing in A4 format. Please take, to make it readable (for example, Fig. 3 and Fig. 4 are perfectly readable). Morevoer, in the inset of panel (h) the red line is missing (while comparing with (b) and (f)) - what is the reason for that? While all these comments are addressed, I will be glad to recommend the publication.
Relpy: (A) It is well known that out-of-time-ordered correlators (OTOCs) grow exponentially in chaotic systems.
To clearly illustrate the exponential growth of the QFI in such systems, it is useful to express the QFI in terms of the fidelity susceptibility, and further rewrite the fidelity using the form of OTOCs. (B) Figure 2 has been revised to improve readability. The absence of the red line in the inset of panel (h) arises from the fact that the CSE model lacks a classical counterpart, making it impossible to extract a Lyapunov exponent in this case.

---

## Round 2 · List of Changes

1) Figure 2 has been revised to improve readability.\ 2) In Sec. 4, we emphasize the consistency between our theoretical predictions and numerical results, even for small system sizes.\ 3) Additional discussion has been included to clarify the connections between our work and related previous studies.\ 4) All changes are highlighted in red for the convenience of the referees.

---

## Round 3 · Author Response

Reply to the Report of Referee A:
Report
I thank the authors for the corrections made. I may again suggest to apply various different chaos measures and compare them directly. This would make the message of this paper much clearer. Possible measures could be entropy growth, spectral quantum chaos measures (distribution of ratios of gaps, higher-order spectral correlation functions), correlations functions, ..., please see any book on quantum chaos.

Reply: We thank the Referee for the interesting suggestion. However, our manuscript is focused on the analysis of the quantum Fisher information. We believe that extending the analysis to entropy growth, spectral quantum chaos measures, and correlation functions-although highly valuable-would blur the focus of our work. We believe that the message of our work is quite clear: unlike regular, stable, as well as unstable dynamics, the chaotic dynamics provide a quantum Fisher information (a quantity related to entanglement and optimal sensitivity in quantum sensing) that is constant in time. The constant value reached after a transient Ehrenfest time is equal to N^2/3, which characterizes Heisenberg scaling, and is recovered by an analytical method. We believe, supported also by the other positive reviewer, that these results are of high interest to a broad community to guarantee publication in SciPost Physics.

Reply to the Report of Referee B:
Reply: We thank the Referee for patiently explaining his/her point of view. However, we respectfully disagree with this interpretation. The misunderstanding arises from the notion of averaging. In our setting, we consider the state $|\Psi_{\rm chaos}(t)\rangle$ generated by chaotic dynamics and then calculate its quantum Fisher information (QFI), $F_Q[|\Psi_{\rm chaos}(t)\rangle;J_\alpha]$, with respect to different measurement directions $\alpha = x,y,z$. In Fig.6 (a-d), we plot the evolution of $F_Q(t)$ under chaotic dynamics [panels (a-c)] and integrable dynamics [panel (d)], considering different measurement directions $\alpha$. Our first main result shows that, in all cases of collective chaotic spin dynamics, the long-time averaged QFI approaches N2/3, i.e.,
\begin{equation}
\lim_{T \to \infty} \frac{1}{T - t^*} \int_{t^*}^T dt \,
F_Q[|\psi_{\mathrm{chaos}}(t)\rangle, \hat{J}_\alpha]
= \frac{N^2}{3},
\end{equation}

where $t^*$ denotes the Ehrenfest time. Equation (1) implies that for $t > t^*$

\begin{equation}
F_Q[|\psi_{\mathrm{chaos}}(t)\rangle, \hat{J}_\alpha]
= \frac{N^2}{3} + \mathcal{O}(N),
\end{equation}

where $\mathcal{O}(N)$ represents the deviation from the dominant term,
i.e., the averaged QFI $N^2/3$. For example, it is obviously shown in Fig.~1(a) when $t > 10$

\begin{equation}
\frac{3 F_Q[|\psi_{\mathrm{chaos}}(t)\rangle, \hat{J}_\alpha]}{N^2} \simeq 1.
\end{equation}

Equation~(2) thus clarifies the meaning of the averaging in Eq.~(1).
In contrast, the Referee’s example involves averaging over a mixture of different pure states.
Indeed, it is generally true that

\begin{equation}
\sum_{k} p_k F_Q[|\psi_k\rangle, \hat{O}]
\;\neq\;
F_Q\!\left[\sum_{k} p_k |\psi_k\rangle \langle \psi_k|, \hat{O}\right].
\end{equation}

We emphasize, however, that our averaging is performed
\textbf{over time, not over ensembles of different quantum states}.

Secondly, since Eq.~(2) is independent of the measurement direction
$\alpha = x, y, z$, we conclude that, compared with the GHZ state, chaotic states
offer an advantage in that no specific measurement direction is required.
This conclusion is supported not only by our analytic derivations but also
by numerical simulations: as shown in Fig.~6(a), the three curves corresponding
to different directions converge to the same value for $t > t^*$.
By contrast, in the integrable model, this behavior is absent, as illustrated in Fig.~6(d).

---

## Round 3 · List of Changes

Changed parts are marked in red.

---

## Editorial Decision

published